# A novel lineage of the *Capra* genus discovered in the Taurus Mountains of Turkey using ancient genomics

Kevin G Daly[1]*, Benjamin S Arbuckle[2], Conor Rossi[1], Valeria Mattiangeli[1], Phoebe A Lawlor[1], Marjan Mashkour[3,4], Eberhard Sauer[5], Joséphine Lesur[3], Levent Atici[6], Cevdet Merih Erek[7], Daniel G Bradley[1]*

[1]Smurfit Institute of Genetics, Trinity College Dublin, Dublin, Ireland; [2]Department of Anthropology, University of North Carolina at Chapel Hill, Chapel Hill, United States; [3]Centre National de Recherche Scientifique / Muséum national d'Histoire naturelle, Archéozoologie, Archéobotanique, Paris, France; [4]University of Tehran, Bioarchaeology Laboratory, (Central Laboratory), Archaeozoology section, Tehran, Islamic Republic of Iran; [5]School of History, Classics and Archaeology, University of Edinburgh, Edinburgh, United Kingdom; [6]Department of Anthropology, University of Nevada, Las Vegas, Las Vegas, United States; [7]Department of Archeology, Department of Prehistoric Archeology, Faculty of Letters, Ankara Hacı Bayram Veli University, Ankara, Turkey

**Abstract** Direkli Cave, located in the Taurus Mountains of southern Turkey, was occupied by Late Epipaleolithic hunters-gatherers for the seasonal hunting and processing of game including large numbers of wild goats. We report genomic data from new and published *Capra* specimens from Direkli Cave and, supplemented with historic genomes from multiple *Capra* species, find a novel lineage best represented by a ~14,000 year old 2.59 X genome sequenced from specimen Direkli4. This newly discovered *Capra* lineage is a sister clade to the Caucasian tur species (*Capra cylindricornis* and *Capra caucasica*), both now limited to the Caucasus region. We identify genomic regions introgressed in domestic goats with high affinity to Direkli4, and find that West Eurasian domestic goats in the past, but not those today, appear enriched for Direkli4-specific alleles at a genome-wide level. This forgotten 'Taurasian tur' likely survived Late Pleistocene climatic change in a Taurus Mountain refuge and its genomic fate is unknown.

*For correspondence:
dalyk1@tcd.ie (KGD);
dbradley@tcd.ie (DGB)

**Competing interest:** The authors declare that no competing interests exist.

## Introduction

The genus *Capra* includes the domestic goat (*Capra hircus*) as well as a variety of wild mountain-dwelling goat/ibex species distributed across Eurasia and North Africa including several listed as endangered or vulnerable (*Pidancier et al., 2006*; *Shackleton, 1997*). Nine species are currently recognized by the IUCN; however, taxonomic relationships are still under revision (*Pidancier et al., 2006*; *Zheng et al., 2020*). Among these, the status of the two species endemic to the Caucasus Mountains has been debated (*Groves and Grubb, 2011*; *Parrini et al., 2009*). The East Caucasian tur (*Capra cylindricornis*) has been considered either a species distinct from the West Caucasian tur (*Capra caucasica*) or they comprise a single species of two potentially-hybridizing populations (*Heptner et al., 1961*). Moreover the bezoar (*Capra aegagrus*), progenitor of domestic goat, has also been reported to hybridize with both tur varieties with which it shares seasonal grazing territories in the Caucasus region (*Pfitzenmayer, 1915*; *Sarkisov, 1953*; *Weinberg, 2002*). Interspecies *Capra* gene flow is well known (*Kazanskaia et al., 2007*; *Manceau et al., 1999*; *Pidancier et al., 2006*),

and may explain discordant phylogenies across loci (*Pidancier et al., 2006*; *Ropiquet and Hassanin, 2006*). Such admixture may have shaped the evolution of domestic goat; for example, the tur has been identified as a putative source of a *MUC6* allele driven to fixation in domestic populations and likely selected for gastrointestinal parasite resistance (*Grossen et al., 2020*; *Zheng et al., 2020*). Tur additionally shows differing affinity to domestic and wild goat genomes indicating a complex evolutionary history of the genus.

Although tur are currently restricted to the Caucasus region, ancient wild goat specimens recovered from Direkli Cave, a camp site used by Late Pleistocene hunters in the Central Taurus Mountains of southern Turkey (*Figure 1A*, *Figure 1—figure supplements 1 and 2*; *Arbuckle and Erek, 2012*), were found to carry a tur-like mitochondrial lineage, designated T (*Daly et al., 2018*). Three of these four reported ancient *Capra* specimen fall within the bezoar autosomal diversity, but a fourth - Direkli4, dated to 12,164–11,864 cal BCE (*Bronk Ramsey, 2009*; *Reimer et al., 2020*) and sequenced here to 2.59 X mean genome coverage (*Table 1*) - shows an excess of ancestral alleles in *D* statistic tests (*Green et al., 2010*) when paired with domestic/bezoar goat (*Figure 1—figure supplement 3*, *Supplementary file 1B*) implying Direkli4 carries ancestry basal to that clade. To explore this signal further, we generated low coverage genomes from historic and rare *Capra* samples, including a 20th century CE zoo-born East Caucasian tur (Tur2), tur specimens from the Dariali-Tamara Fort archaeological site near Kazbegi, Georgia (*Mashkour et al., 2020*), a zoo-born Walia ibex, and supplemented with published modern and ancient *Capra* genomes (*Supplementary file 1A and C*, *Figure 1—figure supplement 4*; *Grossen et al., 2020*; *Zheng et al., 2020*). Surprisingly, a neighbour joining tree from nuclear genome identity-by-state (IBS) information places Direkli4 as sister to a clade of both Caucasian tur taxa, a signal obtained using either goat- or sheep-aligned data (*Figure 1B* and *Figure 1— figure supplement 5*). The Direkli4 genome thus suggests a previously-unrecognized *Capra* lineage sister to both Caucasian tur inhabited the Taurus Mountains ~14,000 years ago.

## Results

Our additional screening of Direkli Cave *Capra* remains identified seven with surviving DNA (*Supplementary file 1D*), with two genomes showing greater affinity to Direkli4 than bezoar from the same site (*Figure 2A*, *Supplementary file 1E*). An MDS plot of IBS distances (*Figure 1—figure supplement 6*) places two Direkli samples with sufficient coverage (Direkli4 and Direkli16) close to East and West Caucasian tur genome clusters, with a slight bias to the former. This tur affinity is unlikely to be driven by error as Direkli specimens have low error rates (0.026–0.195%, *Supplementary file 1A and C*) and do not show inflated distance-to-the-outgroup relative to modern genomes (*Figure 1—figure supplement 7*). A total of 3 out of the 11 Direkli Cave *Capra* specimens therefore are assigned to the tur-related clade, implying that while less numerous than bezoar, members of this clade were not rare in the region in the Late Pleistocene. Nuclear genome types (tur-like or bezoar) do not necessarily co-associate with mitochondrial lineages (tur T and bezoar F), with all combinations except 'tur-like genome, tur-like mitochondria' observed (*Figure 2B*, *Figure 2—figure supplements 1 and 2*, *Supplementary file 1A and E*), establishing that there was gene flow between these lineages. Additionally, there is little variation among Direkli T mtDNA (average 4.07 pairwise differences, compared to 67 among Direkli F mtDNA), suggesting a limited population size for this Direkli tur-like matrilineage.

A tur-like population in the Taurus Mountains is consistent with the high variability in body size of the *Capra* material at Direkli Cave where extremely large *Capra* remains have been reported alongside smaller bezoar-size individuals (*Figure 2—figure supplement 3*). Extant tur exhibit body weights 20–50% larger than bezoar (*Castelló et al., 2016*; *Masseti, 2009*) and it is plausible that the 'large' *Capra* from Direkli represent tur-lineage animals. Although there are clear differences between bezoar and tur horn morphologies (*Pidancier et al., 2006*), unfortunately, diagnostic horncore remains have not been recovered from Direkli. The cave was initially inferred to be occupied primarily during summer months (*Arbuckle and Erek, 2012*), with subsequent discoveries of architectural remains and zooarchaeological analyses indicating more intensive use (*Arbuckle, 2019*). The presence of tur-related goats may reflect use of the cave in the winter months when, based on Caucasian analogs (*Gavashelishvili, 2009*), tur would be expected to descend from the higher elevations surrounding the cave (>2000 m above sea level).

Given the Direkli4 genome was recovered together with bezoar specimens, the two lineages of *Capra* likely had proximate ranges and hybridised. We use *D* statistics (*Green et al., 2010*) to measure

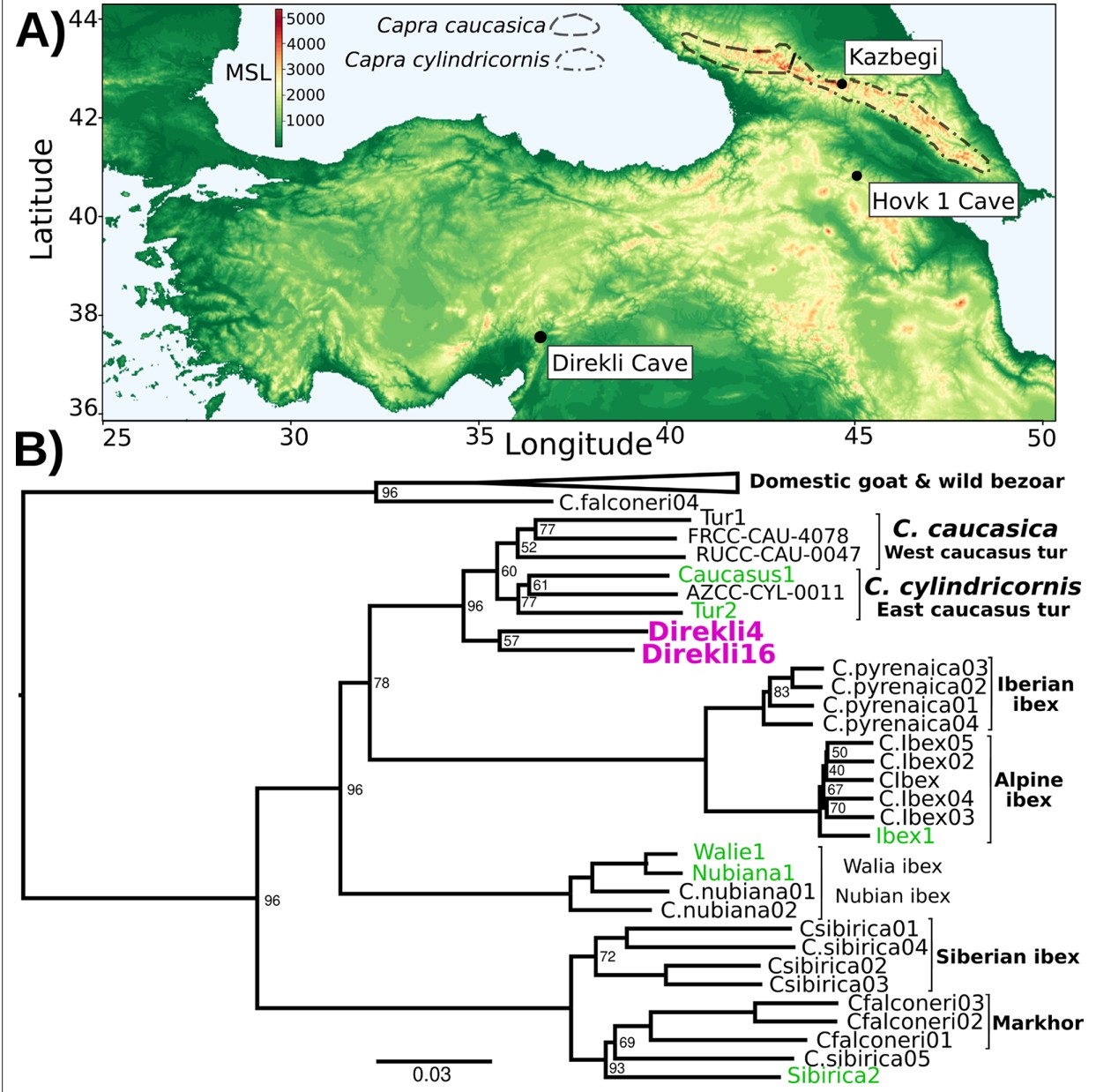

**Figure 1.** Direkli Cave caprids in geographic and genetic context.

(**A**) Elevation map of southwest Asia. Key sites are indicated, with *C. caucasica* and *C. cylindricornis* distributions from *Gavashelishvili et al., 2018* displayed. MSL = metres above mean sea level. (**B**) Neighbour joining phylogeny of genomes >0.5 X and the lower coverage Tur2 (0.02 X) and Direkli16 (0.01 X) genomes using 625,495 transversion sites and pairwise IBS, rooted on Sheep (not shown, as well as a likely Barbary Sheep sample Falconeri1, see Methods). Node support from 100 replicates using 50 5 Mb regions sampled without replacement shown when <100. Pink = Direkli4, green = other genomes first reported here.

The online version of this article includes the following figure supplement(s) for figure 1:

**Figure supplement 1.** View of excavation area from SW, Direkli Cave (from Direkli Cave Excavation Archive, 2018).

**Figure supplement 2.** Plan map of Direkli Cave showing location of *Capra* specimens sequenced in this study.

**Figure supplement 3.** Probability density distribution of pairwise *D* statistics using 28 domestic *C*.

**Figure supplement 4.** Substitution rates of C>T and G>A transitions for ancient and historic samples sequenced in the present study, relative to the 5' and 3' ends of DNA fragments.

**Figure supplement 5.** NJ Phylogeny using sheep-aligned identity-by-state data, (**A**) with and (**B**) without the lower coverage Direkli16 sample.

*Figure 1 continued on next page*

Figure 1 continued

**Figure supplement 6.** MDS plot of sheep-aligned IBS data (**A**) underlying *Figure 1—figure supplement 5* and (**B**) using ancients & historic samples only.

**Figure supplement 7.** Distance-from-the-outgroup for modern and ancient/historic *Capra* genomes.

Direkli4 derived allele sharing relative to either a likely-hunted (*Supplementary file 1G*) or likely-herded (*Supplementary file 1H*, Pearson's *r*>0.99)~10,000 year old goat from the Zagros Mountains. A Late Pleistocene wild goat from the Armenian Lesser Caucasus, Hovk1, shows highest affinity with Direkli4 (*Figure 2—figure supplements 4 and 5*). Bezoar goats from Direkli Cave also show high Direkli4 allele sharing, mirroring affinity measures with west Caucasian tur (*Zheng et al., 2020*). While directionality is uncertain, these statistics imply gene flow between the tur-like lineage and wild bezoar.

Examining domestic goats we find that Neolithic genomes from Europe show greater affinity to Direkli4 (*Figure 2—figure supplement 4*), but Neolithic Iranian goats do not, echoing the distribution of Direkli bezoar-related ancestry in West Eurasian populations (*Daly et al., 2018*). We account for possible gene flow from Caucasian tur into modern European goats using the statistic $D$(Tur1, Direkli4; X, Sheep) to compare relative affinity with Tur1 and Direkli4 (*Supplementary file 1I*). With the exception of two other tur samples, all examined domestic/bezoar goats show either a bias towards Direkli4 or gave a non-significant result, consistent with Direkli4-related admixture or a more complicated genetic history.

Genetic exchange between bezoar and the ancestors of Direkli4 could confound these measures of shared variation among domestic populations. We identified variants specific to Direkli4, conditioned on ancestral allele fixation in a range of defined groups (*Figure 3A*, see Materials and methods). Using this we calculate a statistic analogous to the $D$ statistic, here termed the extended $D$ or $D_{ex}$. $D_{ex}$ measures the relative degree of allele sharing, derived specifically in a selected genome or group of genomes, and may have some utility in genera with complex admixture histories or admixture from ghost lineages. Relative to Neolithic Zagros goats, ancient domestic genomes from western Eurasia have an excess of Direkli4-specific variants (*Figure 3B*, *Figure 3—figure supplement 1*, *Supplementary file 1J*). This 'Direkli4-specific' allele sharing signal is absent in ancient goats from Iran-eastwards, and in all tested modern goat (*Figure 3C*). To control for possible reference biases, we calculated $D_{ex}$ ascertaining on variants segregating in sheep (*Supplementary file 1K*) and recovered similar results (Pearson's *r*=0.9935). Repeating the analysis using other ancient/historic *Capra* 'specific' alleles shows somewhat correlated results (*Supplementary file 1L*), but the distinct patterns of allele sharing (*Figure 3—figure supplements 2 and 3*, *Supplementary file 1M*) imply that Direkli4 ancestry in domestic goat varies temporally and geographically.

We next identify alleles derived in Direkli4 also at a low frequency (>0%,≤10%) in other *Capra* and bezoar, and then measure their abundance in domestic goats. The west Caucasian tur (Tur1) most frequently shares derived alleles with Direkli4 and domestic goat (*Figure 3—figure supplement 4*), consistent with their cladal relationship (although this measure is sensitive to genome depth, see Methods). Ancient European domestic goats share a higher proportion of alleles with both Direkli4 and the high-coverage bezoar from Direkli Cave, Direkli1-2. In comparison, Modern European and African goats carry variation present in Direkli4 plus one of the two Caucasian tur (Tur1 and Caucasus1). This discrepancy could be explained by either gene flow from domestic goats into tur during the last 8000 years, or alternatively an increase in tur-Direkli4 related ancestry in European populations over time.

Investigating gene flow events within *Capra*, automated tree-based model exploration (*Pickrell and Pritchard, 2012*) detects admixture between the Direkli4/Tur lineage and the ancestors of the Late Pleistocene bezoar Hovk-1 (*Figure 3—figure supplement 5*). Residuals of this graph point to unmodeled affinity between Direkli4 and both Direkli Cave bezoar and with Neolithic Serbian domestic goat (*Figure 3—figure supplement 6*). Modern European goat do not show unmodeled Direkli4 affinity, supporting the interpretation that Direkli4-related ancestry has declined with time in west Eurasian goats. A reduced set of populations explored using ML network orientation (*Molloy et al., 2021*) reiterates the Tur1/Direkli4 and Direkli bezoar lineages admixture, and also between Direkli bezoar and domestic goat (*Figure 3—figure supplement 7*). Investigating admixture graph space (see Materials and methods, *Maier et al., 2022*) we find 2 admixture events best explain how a

**Table 1.** Sample provenance and sequencing summary.

| Sample | Morphological Species | Origin | Age | Sex† | Nuclear Cov. ‡ | mtDNA Cov. |
|---|---|---|---|---|---|---|
| Direkli4 | Capra spc. | E4/8 A, Direkli Cave, Turkey | 12,164–11,864 cal BCE (2σ) | M | 2.59 | 642.23 |
| Direkli9 | Capra spc. | B6/5, Direkli Cave, Turkey | Est. 12,100–8,900 BCE* | M | 0.0003 | 9.5 |
| Direkli12 | Capra spc. | B13/4 A, Direkli Cave, Turkey | Est. 12,100–8,900 BCE | M | 0.07 | 44.06 |
| Direki13 | Capra spc. | B13/7B, Direkli Cave, Turkey | Est. 12,100–8,900 BCE | M | 0.0055 | 76.99 |
| Direkli14 | Capra spc. | D3/7, Direkli Cave, Turkey | Est. 12,100–8,900 BCE | M | 0.0005 | 13.24 |
| Direkli15 | Capra spc. | B6/5, Direkli Cave, Turkey | Est. 12,100–8,900 BCE | F | 0.0002 | 24.21 |
| Direkli16 | Capra spc. | B8/7, Direkli Cave, Turkey | Est. 12,100–8,900 BCE | F | 0.01 | 15.11 |
| Direkli17 | Capra spc. | E5/5, Direkli Cave, Turkey | Est. 12,100–8,900 BCE | F | 0.0001 | 0.0845 |
| Caucasus1 | Capra caucasica | Tamara Fort, Kazbegi, Georgia | 4th-21st c. CE, probably 5th-15th c. CE | M | 0.55 | 64.63 |
| Caucasus2 | Capra caucasica | Tamara Fort, Kazbegi, Georgia | 4th-21st c. CE, probably 5th-15th c. CE | M | 0.0021 | 379.71 |
| Caucasus3 | Capra caucasica | Tamara Fort, Kazbegi, Georgia | 4th-21st c. CE, probably 5th-15th c. CE | M | 0.004 | 464.51 |
| Falconeri1 | Capra falconeri hepteneri § | Unknown, via Parc de la Haute-Touche | 20th Century CE | M | 0.58 | 72.11 |
| Falconeri2 | Capra falconeri | Born at MNHN Zoo, Paris | 20th Century CE | M | 0.06 | 45.78 |
| Ibex1 | Capra ibex | Unknown | 20th Century CE | F | 3.93 | 179.03 |
| Ibex2 | Capra ibex | Pointe de Calabre, Savoie | 20th Century CE | M | 0.05 | 21.4 |
| Sibirica1 | Capra sibirica | Born at MNHN Zoo, Paris | 20th Century CE | M | 0.04 | 163.16 |
| Sibirica2 | Capra sibirica | Born at MNHN Zoo, Paris | 20th Century CE | F | 1.48 | 205.78 |
| Tur2 | Capra cylindricornis | Unknown, via Vincennes Zoo | 20th Century CE | F | 0.02 | 4.96 |
| Walie1 | Capra walie | Born at MNHN Zoo, Paris | 20th Century CE | M | 0.75 | 103.12 |
| Pyrenaica2 | Capra pyrenaica | Unknown | 20th Century CE | M | 0.16 | 51.2 |
| Nubiana1 | Capra nubiana | Unknown | 20th Century CE | M | 1.25 | 211.32 |

*Estimated ages for Direkli material is based on calibrated ages from the cave stratigraphy.

†M=Male, F=Female.

‡Cov. = coverage.

§Falconeri1, a likely Barbary sheep (see Materials and methods).

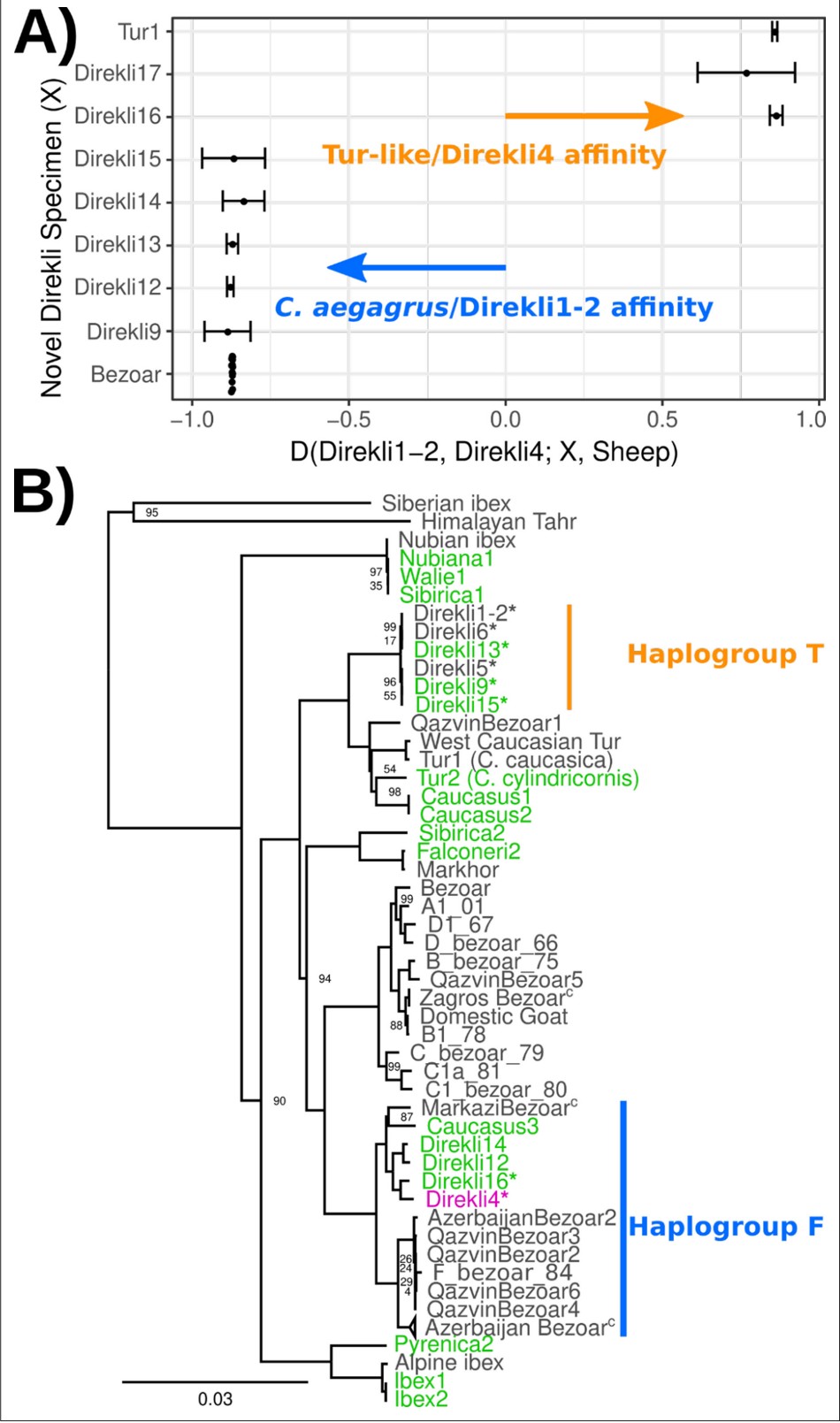

**Figure 2.** Autosomal affinity and mtDNA profile of Direkli Cave caprids. (**A**) *D* statistic test of affinity using specimens from Direkli Cave and a historic *C.caucasica* individual for reference. Positive values indicate sample X has greater affinity with the Tur-like Direkli4 genome; negative values indicate greater affinity with the *C. aegagrus* Direkli1-2. Error bars represent 3 standard errors, underlying site counts are presented in ***Supplementary file***

*Figure 2 continued on next page*

*Figure 2 continued*

*1E*. (**B**) ML phylogeny of mtDNA, abbreviated. Bootstrap node support values (100 replicates) are displayed when <100. The complete phylogeny including likely Barbary sheep Falconeri1 is displayed in *Figure 2—figure supplement 1*. T haplogroup is as defined by *Daly et al., 2018* Low coverage sample Direkli17 is displayed in a highly reduced phylogeny *Figure 2—figure supplement 2*. C=collapsed. *=Direkli sample with discordant mtDNA and nuclear genome affinity.

The online version of this article includes the following figure supplement(s) for figure 2:

**Figure supplement 1.** ML phylogeny of mtDNA, uncollapsed.

**Figure supplement 2.** Reduced mtDNA ML phylogeny including Direkli17.

**Figure supplement 3.** *Capra* body size estimates from archaeological assemblages, organized by region, time, and archaeological site.

**Figure supplement 4.** *D* statistic of the form *D*(Abdul4, Test; Direkli4, Sheep), where Abdul4 is a *Capra aegagrus* from the 11th millennium BCE Zagros Mountains and Direkli4 is a *Capra caucasica* like genome from the Taurus Mountains.

**Figure supplement 5.** *D* statistic of the form *D*(Abdul4, Test; Direkli4, Sheep), where Abdul4 is a *Capra aegagrus* from the 11th millennium BCE Zagros Mountains and Direkli4 is a *Capra caucasica* like genome from the Taurus Mountains.

subset of populations (Sheep, Tur1, Direkli4, Direkli bezoar, Neolithic East Iran, and Neolithic Serbia) can be modeled. A majority (6/11) of graphs model Direkli bezoar as containing ancestry related to Direkli4 (median 1.5%, mean 5.2%; best fitting graph is shown in *Figure 3—figure supplement 8*), with a single graph modeling the opposite (2% Direkli bezoar ancestry). While the graph space explored is limited, these results suggest a greater degree of 'Direkli4 to Direkli bezoar' gene flow than 'Direkli bezoar to Direkli4'.

We finally identify 3 out of 112 regions introgressed from other *Capra* species to domestic goat (*Zheng et al., 2020*) which show high affinity with Direkli4 (*Figure 3—figure supplements 9–11*, 21–22). A further nine regions appear to have most affinity with the Direkli4-tur clade (*Figure 3—figure supplements 12–20*), including a locus encompassing *MUC6*, a target of selection in domestic goats during the last 10,000 years (*Zheng et al., 2020*), implicating the Direkli4 lineage in the makeup of domestic goat gene pool.

## Discussion

Our results indicate that a lineage related to the Caucasian tur existed in the Taurus Mountains during the Late Pleistocene, as late as the 12th millennium cal BCE. Based on the current, limited genomic data from the *Capra* genus, which we improve on here, this lineage appears to be a sister group to the tur *C. caucasica* and *C. cylindricornis*. Similar to other mammalian groups (*Gopalakrishnan et al., 2018*; *Palkopoulou et al., 2018*; *Zheng et al., 2020*), admixture likely occurred among *Capra* lineages; the population reported here carries bezoar-associated mtDNA and a possible small amount of bezoar nuclear genome ancestry (2% from 1/12 graphs). This Taurasian tur population is itself a possible candidate for the source of Tur-like ancestry present in domestic goats, including an introgressed *MUC6* allele fixed in modern populations which increases gastrointestinal parasite resistance (*Zheng et al., 2020*). Given the relative paucity of *Capra* genomic data available compared to other mammalian groups, additional genomes from the genus will help refine the history of divergences and gene flow events which shaped the group's evolution.

We suggest this novel 'Taurasian tur' lineage be designated *Capra taurensis* following IUCN convention (*Weinberg and Lortkipanidze, 2020*) or *Capra caucasica taurensis* under a subspecies classification (*De Queiroz, 2007*; *Wilson and Reeder, 2005*). The Taurasian tur may have diverged from the Caucasian lineages 130-200kya based on mtDNA coalescent estimates (*Bouckaert et al., 2014*; *Daly et al., 2018*). The current distribution of *Capra* species is mostly discontinuous and is suggestive of climate-induced fragmentation (*Shackleton, 1997*). The ancestors of Caucasian tur likely extended over a broader range in Eurasia during the Late Pleistocene but may have been poorly captured by the fossil record (*Crégut-Bonnoure, 1991*; *Uerpmann, 1987*; *Weinberg, 2002*). The large variability and high upper size range of *Capra* remains are consistent with both smaller-bodied bezoar and larger-bodied tur-relatives being present within the faunal assemblage at Direkli as well as other sites in the

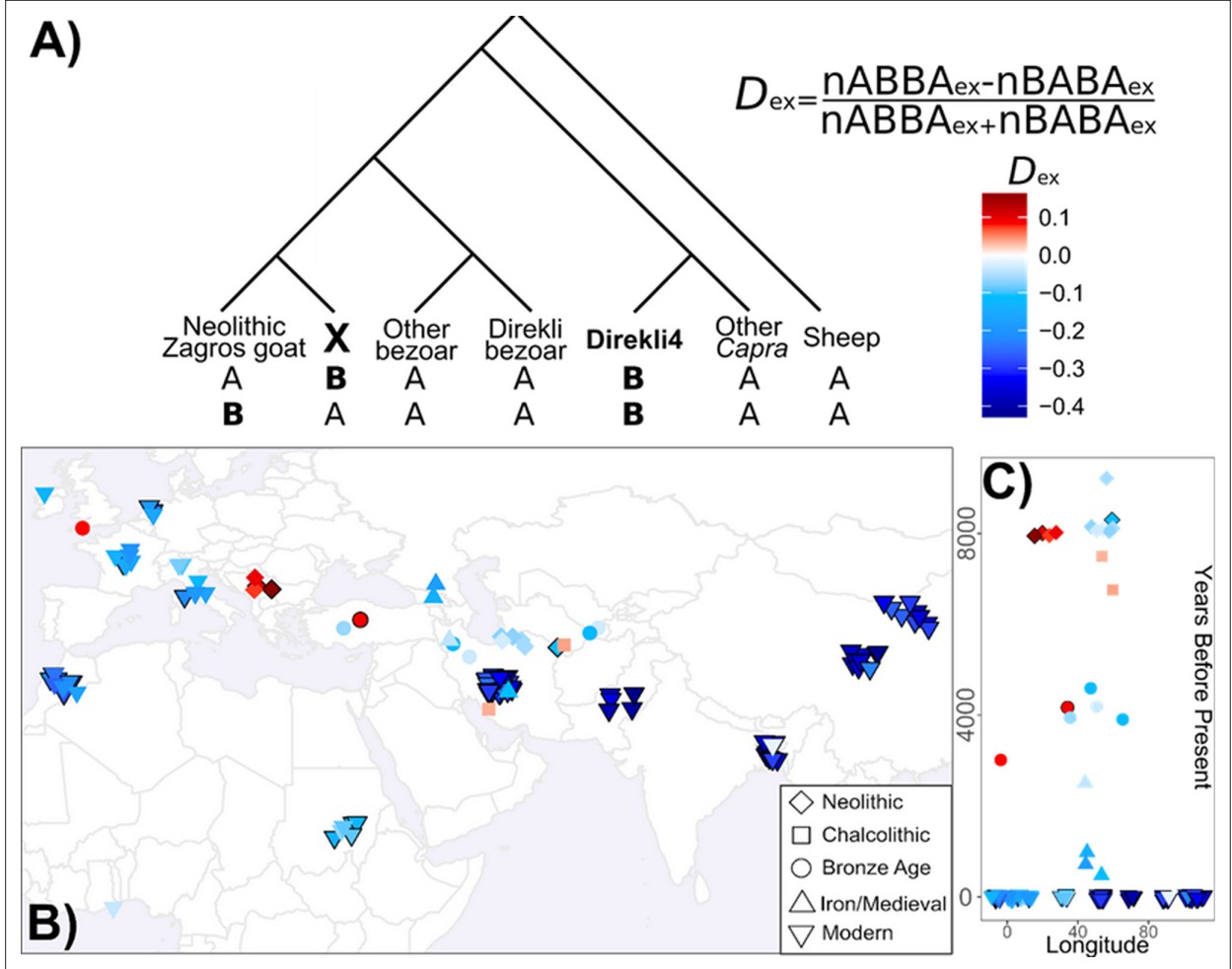

**Figure 3.** Extended D calculation and application to Direkli4 specific allele sharing. (**A**) Extended *D* statistic. To control for gene flow with the *Capra* genus, we condition on variants derived in Direkli4 (H3) and a genome X (H2), but ancestral in other populations (here: Sheep, non-bezoar *Capra* genomes, bezoar from Direkli Cave, and other bezoar). Values are calculated relative to a set of Neolithic goat from the Zagros Mountains, and normalized similarly to the *D* statistic. (**B**) Extended D statistic values for Direkli4 using transversions, (**C**) plotted through time. Each symbol is a test genome, with shape denoting time period. Black borders indicates a |Z| score ≥3, using 1000 bootstrap replicates and 5 Mb blocks.

The online version of this article includes the following figure supplement(s) for figure 3:

**Figure supplement 1.** Extended D statistic in the form {Aceramic Neolithic Zagros, X/H2, Direkli4 (H3), Direkli Wild + Other Wild + Capra, Sheep}, using both transitions and transversions.

**Figure supplement 2.** Extended *D*, rotating H3 through *Capra* genus genomes, first tranche of H3 genomes.

**Figure supplement 3.** Extended *D*, rotating H3 through *Capra* genus genomes, second tranche of H3 genomes.

**Figure supplement 4.** Distribution of Direkli4-specific variants also present at >0%,≤10% in wild genomes, expressed as a proportion of the total number of these variants in the tested domestic genomes.

**Figure supplement 5.** Treemix results for *m*=5, using 223,772 transversion variants ascertained on sheep, k=1000.

**Figure supplement 6.** Residuals for Treemix m=5.

**Figure supplement 7.** Orientagraph model and residuals for m=4, using 104,550 transversion variants and k=1000.

**Figure supplement 8.** ADMIXTOOLS2.

**Figure supplement 9.** Clustered heatmap of IBS data for ARS1 1:104,150,173–104,349,720.

**Figure supplement 10.** Clustered heatmap of IBS data for ARS1 2:24,324,410–24,369,675.

**Figure supplement 11.** Clustered heatmap of IBS data for ARS1 13:66,710,508–66,749,824.

**Figure supplement 12.** Clustered heatmap of IBS data for ARS1 1:118,433,695–118,469,862.

**Figure supplement 13.** Clustered heatmap of IBS data for ARS1 6:20,670,803–20,728,594.

*Figure 3 continued on next page*

central Taurus and Lebanese mountains (Üçagızlı cave, Ksar Akil, Saaide II), but not in the western Taurus where only bezoar are evident (*Figure 2—figure supplement 3*; *Arbuckle and Erek, 2012*). *C. taurensis* could have survived the Last Glacial Maximum within the central Taurus Mountains, a plausible refugia for *Capra* species in addition to the Pontic and Anti-Taurus ranges (*Gavashelishvili et al., 2018*) while experiencing a severe matrilineal bottleneck (*Figure 2B*). *C. taurensis* appears to have produced fertile offspring with other members of the *Capra* genus; the traces of shared ancestry in ancient bezoar and likely managed goat (*Figure 3B*) may be the consequence of direct gene flow or secondarily via admixed bezoar. Gene flow between early managed goats and Anatolian bezoar carrying *C. taurensis* ancestry could partially explain the divergence between Zagros and more westerly herds.

Given the tremendous pressure on *Capra* species via Anthropocene over-hunting and habitat disruption (*Shackleton, 1997*), it is assumed that *C. taurensis* is extinct, with its existence only now revealed via palaeogenomics. The Caucasian tur's preference for snowier habitats (*Gavashelishvili et al., 2018*) combined with the lower altitude of the Taurus Mountains relative to the Caucasus (*Figure 1A*) may have rendered the lineage vulnerable to climatic change via Holocene warming and interspecific competition with bezoar, which are still found in the Taurus mountains (*Gavashelishvili, 2009*; *Naderi et al., 2008*), leading to its hypothesised extinction. As the history of *C. taurensis* following the Late Pleistocene is still unknown, further genomic surveys of Holocene *Capra* remains and present-day populations, such as the VarGoats project (*Denoyelle et al., 2021*), from this and adjacent regions may illuminate its genetic legacy.

## Materials and methods
### Materials and methods summary
DNA from 7 postcranial bone elements from Direkli Cave and 13 historic Capra specimen was extracted via standard aDNA protocols (*Yang et al., 1998*), with a 0.5% sodium hypochlorite pre-wash (*Korlević et al., 2015*) performed for the Direkli material. Following uracil excision (*Rohland et al., 2015*) and dsDNA library construction (*Meyer and Kircher, 2010*), libraries were subject to shotgun sequencing (Illumina HiSeq 2000 and NovaSeq 6000) or RNA-bait enrichment of mtDNA reads prior to shotgun sequencing. Additional sequencing data was also generated for specimen Direkli4.

Using bwa aln (*Li et al., 2008*) a relaxed alignment (*Meyer et al., 2012*) was performed against the goat reference ARS1 (*Bickhart et al., 2017*) or an outgroup genome (Oar_rambouillet_v1.0). Subsequent analyses were primarily performed in the ANGSD environment (*Korneliussen et al., 2014*) using single read sampling.

### Direkli Cave
Direkli Cave (37°51'28.08"N, 36°39'13.98"E) is located in the Central Taurus mountains, north-west of Kahramanmaraş Province, near the village of Döngel, Turkey (*Erek, 2010*; *Erek, 2012*). It was discovered and first excavated by *Kökten, 1960*. The cave sits in a south-facing limestone escarpment at an elevation of 1100 m above sea level. It is located at the base of the Deli Höbek mountain and adjacent peaks which rise up to 2200 m immediately to the northeast. The cave is located above a small alluvial fan on the eastern side of the north-south trending valley draining the Tekir river (and

the D825 highway), itself a tributary to the Ceyhan, a major river system which flows into the Mediterranean. The site is located on the edge of a highly dissected upland and within a corridor providing easy access both to the Mediterranean coast as well as the interior of southern Turkey. It is therefore situated along a convenient route for seasonal movements between higher and lower elevations for both humans and other animal species.

In 2007, new excavations under the direction of C. M. Erek were initiated with the support of the Turkish Ministry of Culture and Tourism in order to further explore the late Pleistocene deposits in the cave, a period that is poorly documented for the central Taurus mountains. Work carried out since that time has identified a stratigraphic sequence rich in artifacts and features dating to the late Pleistocene with lithic parallels both with the Levantine Natufian as well as sites on the Turkish Mediterranean coast. Radiocarbon dates derived from charcoal and animal bone from layers containing microlithic tools, date the main prehistoric occupation layers of the site to between 12,100–8900 years calibrated BCE. The majority of these dates place the occupation in the late Pleistocene prior to the Younger Dryas although the cave was used as late as the late 10th millennium BCE as well.

Excavations were carried out following the natural and cultural stratigraphic layers. Sediments of different colors in each plan square were collected in different buckets and passed through a triple screening system. The sediments, which were all subjected to water screening, were sieved through fine (1 mm), medium (2 mm), and coarse (4 mm) sieves and the materials in each sieve were dried in the shade. These recovery techniques have produced a rich faunal assemblage indicative of the local fauna exploited by the cave's occupants. This faunal community is dominated by the remains of wild goats (*Capra spp*–the focus of this study) but also deer, boar and black bear as well as a variety of fur bearing carnivores.

*Capra* specimens sequenced in this study derive from excavation areas located in the North portion (interior) of the cave (*Figure 1—figure supplement 1*). Stratigraphic affiliation of each specimen is reported in *Table 1*. Specimens were recovered from stratigraphic layers 4–8 and all derive from deposits containing material culture (especially microliths) consistent with a late Pleistocene date. In general, *Capra* specimens are highly fragmented and many exhibit cut and percussion marks clearly indicating an anthropogenic origin for the assemblage. There is no evidence that *Capra* remains accumulated through natural processes or carnivore denning behaviours. Direct AMS dates were acquired for two specimens (Direkli4 [Beta-432464: 12130+/-40] and Direkli2 [Beta-425280: 11370+/-40]) confirming their late Pleistocene provenience (12,200–11,200 cal BCE). Four specimens derive from grid square B/13 (*Figure 1—figure supplement 2*) including stratigraphic levels 4 A, 4 C, 7B, 7 C providing a temporal sequence from youngest to oldest including Direkli2, Direkli15, Direkli16, Direkli1.

## Sample preparation

Material from the Muséum national d'Histoire naturelle collections in Paris were sampled on-site. Newly screened specimens from Direkli Cave and Dariali-Tamara Fort were sampled in dedicated ancient DNA facilities in TCD, Dublin, following standard protocols (*Pääbo et al., 2004*).

Sampled materials were cleaned with a drill bit and then subject to UV for 30 min, flipping midway to decontaminate both sides, followed by pulverization using a Mixer Mill (MM 200, Retsch).

## DNA extraction and library preparation

For MNHN and Tamara Fort material, ~150 mg of bone powder was subject to EDTA-prewash and proteinase K 3 day extraction as previously described (*Daly et al., 2018*; *Gamba et al., 2014*; *MacHugh et al., 2000*; *Yang et al., 1998*). For newly screened Direkli Cave material, samples were subjected to a 0.5% hypochlorite wash prior to EDTA wash and proteinase K-based extraction (*Boessenkool et al., 2017*; *Daly et al., 2021*). Previously described protocols for DNA cleanup (*Daly et al., 2018*; *Daly et al., 2021*), Uracil-DNA-glycosylase treatment *Rohland et al., 2015* of 16.25 µl DNA followed by dsDNA library construction (*Meyer and Kircher, 2010*). Control tubes were included for extraction and library steps and kept for subsequent analysis, but did not show evidence of contamination.

## Screening, sequencing and nuclear genome alignment

Libraries were amplified (Accuprime Pfx, Thermofisher) with single indexes (MNHN and Tamara Fort specimen) or double indexes (newly-screened Direkli specimen) as described (*Daly et al., 2018*; *Daly*

*et al., 2021*). Initial screening was performed using an Illumina MiSeq platform (50 bp SE; TrinSeq, Dublin), HiSeq 2000 (100 bp SE; Macrogen, Seoul) or NovaSeq 6000 (100bp PE; TrinSeq, Dublin); sample-platform mix is presented in *Supplementary file 1D*.

Following de-multiplexing, single-end fastq files were filtered for adaptors which were also trimmed, and reads <30 bp in length removed using cutadapt v1.9.1 (*Martin, 2011*) cutadapt -a AGATCGGA AGAGCACACGTCTGAACTCCAGTCAC -O 1 m 30.

AdaptorRemoval v2.3.1 (*Schubert et al., 2016*) was used for pair-end data, for the above QC steps and also to collapse overlapping read pairs: AdapterRemoval `-collapse --minadapteroverlap 1 --adapter1` AGATCGGAAGAGCACACGTCTGAACTCCAGTCAC `--adapter2` AGATCGGAAGAG CGTCGTGTAGGGAAAGAGTGT `--minlength 30 --gzip --trimns --trimqualities`.

Trimmed (SE) or collapsed (PE) reads were aligned to ARS1 (*Bickhart et al., 2017*) using bwa aln (*Li et al., 2008*) relaxing parameters (*Meyer et al., 2012*) and handling sam/bam files using samtools v1.11 (*Li et al., 2009*). Read groups were assigned using bwa samse/sampe to differentiate sequencing libraries and PCR reactions. To estimate endogenous DNA %age, duplicate removal was performed using picard v2.22.1 (*The Broad Institute, 2018*) and mapQ ≥30 filtering with samtools, dividing QCed aligned reads by raw reads. Substitutions associated with ancient DNA were examined using mapDamage2 (*Jónsson et al., 2013*), and are presented in *Figure 1—figure supplement 4*.

Samples selected for deeper sequencing (typically >10% endogenous DNA, and including additional sequencing of Direkli4 originally reported in *Daly et al., 2018*) were sequenced on either a Illumina HiSeq 2000 (100 bp SE; Macrogen, Seoul) or NovaSeq 6000 (100bp PE; TrinSeq, Dublin). Deeper sequenced data was aligned as above, with a subsequent indel realignment step (*The Broad Institute, 2018*) and softclipping of reads by reducing the base quality of the first and last four bases to zero using a custom python script. Genome depth was estimated using GATK and presented along with alignment statistics in *Supplementary file 1D*.

For alignment to the sheep reference, the above pipeline was used substituting ARS1 with the sheep reference Oar_rambouillet_v1.0 available at https://www.ncbi.nlm.nih.gov/assembly/GCF_002742125.1/. Samples aligned to sheep are indicated in *Supplementary file 1C*.

## Mitochondrial capture
Samples not selected for deep genomic sequencing were enriched for mammalian mtDNA sequences using an in-solution RNA hybridization approach using custom baits as previously described (*Daly et al., 2018*; *Gnirke et al., 2009*; *Maricic et al., 2010*; *O'Sullivan et al., 2016*) Daicel Arbor Biosciences, Ann Arbor, USA. Libraries enriched for mtDNA were subsequently sequenced using a MiSeq platform (50 bp SE; TrinSeq, Dublin).

## Mitochondrial genome alignment and phylogeny
Reads were aligned to circularized (15 bp either end) mitochondrial reference genomes as previously described (*Daly et al., 2021*), realigning to closer mtDNA references to maximize sequence recovery. Sample fasta files were producing using ANGSD v0.922 (-doFasta2 -setMinDepth 3 -minQ 20 -minMapQ 30 -trim 4; *Korneliussen et al., 2014*) and decircularized by removing 15 bp at both ends of the resulting fasta sequence. Due to the low coverage of Direkli17, this sample fasta was produced with -setMinDepth 1 and analyzed independently of others.

A multiple sequence alignment of data generated here plus published *Capra* mtDNA sequences (*Supplementary file 1E*) was generated using MUSCLE (*Edgar, 2004*). A ML phylogeny with 100 bootstrap replicates was then computed using phyML (*Guindon et al., 2010*) using a BIC-selected substitution model, and visualized using Figtree (*Rambaut, 2009*).

Pairwise differences among aligned mtDNA sequences was calculated using a custom Python script, ignoring sites where one of the two samples were missing data ('N'). For F lineage comparisons, Direkli17 was excluded due to its low coverage and the higher error rate implicit in the -setMinDepth 1 option used for that sample.

## Modern alignment
Modern genomes (*Supplementary file 1C*) were aligned to ARS1 or Oar_rambouillet_v1.0 using bwa mem v0.7.5a-r405 (*Li, 2013*; *Li et al., 2009*; *Li et al., 2008*) and following GATK Best Practices, removing duplicates and reads with mapQ ≥30. Among these were 3 modern Tur genomes, which

were included in the IBS calculation and nj tree (below) with permission from the VarGoats consortia (*Denoyelle et al., 2021*) prior to these samples' in-depth analyses. Samples aligned to sheep are indicated in *Supplementary file 1C*.

## ANGSD-based analyses

All ANGSD (v0.922; *Korneliussen et al., 2014*) analyses were performed using the following parameters: -minMapQ 30 -minQ 20 C 50 -remove_bads 1 -uniqueOnly 1 -rmTrans 1 and restricting analyses only to the autosomes. When necessary (-doMajorMinor 5 or -doAbbababa 1), the ancestral sequence was defined using a fasta file from a goat-aligned sheep (*Daly et al., 2021*). Genotype likelihoods were computed using -GL 2 -rmTrans 1 -SNP_pval 1e-6 -skipTriallelic 1.

## Error estimation

To estimate the error rates of the Direkli specimen and *Capra* genomes reported here, the ANGSD 'perfect genome' approach was employed, using a high coverage Old Irish Goat ('IOG',~42 X): -doAncError 1 -ref IOG.fa, where IOG.fa was generated using ANGSD -doFasta 1 -doCounts 1 -setMinDepth 13 -setMaxDepth 76 C 50 -minQ 20 -minMapQ 30. Sheep was used as the outgroup. Error rates were reasonable, with all falling below 0.5% and the highest reaching 0.3428%; the highest Direkli genome error was just 0.1947%. Error rates are shown in *Supplementary file 1A and C* (for previously published Direkli bezoar and the Tur1 specimen).

We additionally assessed error rates of ancient vs modern individuals by plotting distance from the outgroup as calculated from Identity-By-State statistics (see below). We expect high error individuals, particularly ancient ones, to show inflated distances from the outgroup (sheep). Plotting distances (*Figure 1—figure supplement 7*), the majority ancient Direkli specimens do not show excessive distance to the outgroup, with the highest distances being observed in historic and modern European (alpine and Iberian) ibex. A single Direkli wild goat / *Capra aegagrus* shows elevated distance to the outgroup relative to modern *Capra aegagrus* (0.240308), but all other Direkli specimens have unremarkable distance values.

## D statistics

D statistics were computed using -doAbbaBaba 1 -rmTrans 1 -doCounts 1 and using sheep as above to define the ancestral allele. Correlation between *D* statistical tests was measured by calculating Pearson's correlation coefficient *r* using the cor() function of *R Development Core Team, 2021*. *D* test results are presented in *Figure 2—figure supplements 4 and 5*, and *Supplementary file 1GH and I*. A subset of *D* statistic tests were also repeated using sheep aligned-data (see above); Pearson's *r* for the test indicated in *Supplementary file 1FG and H* were 0.927, 0.950, and 0.494, respectively. The latter is not concerning as the *D* statistics in question (Direkli4, Tur1; X, Sheep) are overwhelmingly large for goat and sheep-aligned data (i.e. |D|>>0.05) and are consistent in directionality.

## Identity-by-state (IBS)

Pairwise IBS matrices were computed for reported genomes ≥0.5 X coverage, the lower coverage Tur2 genome (*C. cylindricornis*), a subset of modern and ancient *Capra* genomes, and a sheep outgroup using ANGSD (-makeMatrix 1 -doCov 1 -doMaf 1 -minFreq 0.05 -minInd 56), allowing ~5% missingness per site. Neighbour-joining phylogenies were constructed using the nj() function of the ape R package (*Paradis and Schliep, 2019*). Published genomes included in IBS calculations are indicated in *Supplementary file 1C*; sheep was used as the outgroup.

To calculate node support, pseudo-bootstrap datasets were generated by sampling-without-replacement 50 5 Mb regions a total of 100 times, and calculating IBS values for each of the 100 pseudoreplicates. Neighbour-joining trees were calculated as above and node supports applied to the base tree using RAxML (*Stamatakis, 2014*), and are presented in *Figure 1B*.

To control for possible reference genome effects, an additional IBS neighbour-joining tree was computed using sheep-aligned data using the following ANGSD settings, allowing for 90% site missingness due to the smaller number of genomes used: -doMajorMinor 4 GL 2 -minFreq 0.05 -minInd 41. Node support values were calculated as above, using 50x5 Mb regions per replicate. This tree is presented in *Figure 1—figure supplement 5*, with (A) and without (B) the lower coverage Direkli16

sample which falls within the 'taurasian tur' clade. Site counts for both trees are 96,930 and 110,672 respectively.

We additionally constructed a MDS plot using the sheep-aligned MDS data and R functions cmdscale(as.dist()) on the IBS.ibsMat file (*Figure 1—figure supplement 6A*), with an insert of the corner of the plot containing the taurensis tur lineage. The MDS places the nubian, alpine, and european ibex specimens, as well as tur, in a distinct part of the MDS plot away from a *Capra aegagrus/hircus* group and a *Capra sibirica/falconeri* (Siberian ibex and markhor). Within the west Eurasian ibex & tur genomes, the relative affinities seen in the nj tree (*Figure 1B*) are reiterated, with the tur group being closer to the European ibex than to Nubian ibex. The taurasian tur genomes Direkli4 and Direkli16 clearly fall within the Tur group. Both eastern and western tur genomes form distinct groups in MDS space, with Direkli tur showing somewhat higher affinity to the eastern tur group. We additionally computed IBS using sheep-aligned, ancients & historic samples only, allowing 20% missingness to account for the relatively high number of low coverage samples (-minInd 21). From the resulting IBS matrix (computed from 464,799 biallelic transversion sites), an MDS plot was produced (*Figure 1—figure supplement 6B*). This plot offers less discriminatory power within the 'ibex / tur' group, as expected given the lower number of *Capra* genomes and overall lower coverage among the non *Capra hircus / Capra aegagrus* genomes. However, both Direkli 'taurasian tur' genomes fall clearly within the 'ibex / tur' group.

## Haploid calling

To produce a haploid (random read sampling) callset of genomes with at least 0.5 X coverage for subsequent analyses the ANGSD -doHaplocall function was employed: -doHaploCall 1 -minInd 2 -minQ 20 -minMapQ 30 -remove_bads 1 -uniqueOnly -doCounts 1 C 50 -trim 4. Output files were then screened using a custom python to retain biallelic sites only and remove CpG sites (232,685,198 remaining), and then converted to plink format (*Purcell et al., 2007*) using the haploToPlink tool provided with ANGSD. An additional dataset consisting of genomes ≥2 X coverage was also generated for Treemix and Orientagraph analyses (below).

## Outgroup/sheep ascertained sites

To create an outgroup-ascertained set of variant sites for Treemix/Orientagraph, Minor Allele Frequencies were computed dataset of 11 ARS1-aligned sheep (*Supplementary file 1C*).

-doMaf 1 -doCounts 1 -minMaf 0.05 -minQ 20 -minMapQ 30 C 50 -doMajorMinor 3 GL 2 -remove_bads 1 -uniqueOnly 1 -SNP_pval 1e-6 -setMinDepthInd 4 -minInd 6 -sites sites.txt; where sites.txt was a file describing biallelic sites in the 2 X haploid-called dataset (above). These 223,772 outgroup-varying sites were then extracted from the 2 X dataset using plink --extract. Twelve modern domestic goat samples (Alashan2, 3, and 4; Erlangshan1, 3, and 4; Aerbasi1, 3 and 6; Liaoning2, 3, and 5; see *Supplementary file 1C*) were then removed due to an excess of missingness (>0.15).

## Introgressed regions

To examine possible sources of 112 haplotypes identified as introgressed in domestic goat (*Zheng et al., 2020*), IBS values were calculated using ANGSD (-doIBS 1 -doMajorMinor 5 -minInd 157 -minMaf 0.02). The -minInd parameter was set to ensure only sites with ≤10% missing data across the 175 individuals were considered, and -minMaf so that alleles found at least one *Capra* source and two domestic genomes were included. Genomes with ≥1 X mean coverage were included, plus the *Capra* samples of interest. Introgressed region coordinates were obtained from Data *Supplementary file 1* of *Zheng et al., 2020*.

Heatmaps from pairwise ibs data were constructed using the R gplots heatmap.2() function (*Warnes et al., 2019*), first removing Nubiana1 and Caucasus1 due to coverage effects, and sheep samples due to distortion of ibs distance value scales. Heatmaps and hierarchical clustering of each putatively-introgressed region are displayed in *Figure 3—figure supplements 9–22*. Additionally, nj trees were constructed for each IBS matrix using the nj() function and visually assessed to determine if the Direkli4 lineage is a possible source of introgressed haplogroup. All 10 nj trees are shown in *Figure 3—figure supplements 21 and 22*. Assessing nj trees and heatmaps, 3 regions out of the 112 total are plausible as Direkli4 being the closest-to-domestic sample (1:104,150,173–104,349,720; 2:24,324,410–24,369,675; 13:66,710,508–66,749,824); a further 7 may fit with the Direkli4-Tur1 clade

being closest to domestic. These assignments should be considered preliminary in lieu of higher quality genomic data for the two Caucasian tur and the Direkli4 lineages.

## Identity of MNHN sample Falconeri1

Falconeri1, thought to be a *Capra falconeri hepteneri* (Tadjik or Bukharan markhor) from the Muséum national d'Histoire naturelle (MNHN-ZM-AC-2009–243), shows the genetic profile aberrant relative to its supposed species. The mitochondrial sequence of Falconeri1 groups with the Barbary sheep or aoudad (*Ammotragus lervia*), a member of the *Caprini* tribe but not of the genus *Capra* (*Figure 2—figure supplement 1*). The mtDNA does not form a clade with the two other markhor sequences included in the phylogeny. Consistent with this is the position of Falconeri1 in IBS-based nj trees, which places the sample basal to all other non-outgroup samples, and does not group with other markhor samples (*Figure 1B*, *Figure 1—figure supplement 5*). We conclude that the sample was misidentified during storage or sampling, and that the genetic sample labeled here as Falconeri1 is more likely to be a Barbary sheep. As such the sample was excluded from subsequent analyses.

## Extended *D*/Direkli4-specific alleles

We extended the general idea of the *D* statistic (*Green et al., 2010*; *Patterson et al., 2012*) and group *D* statistic (*Soraggi et al., 2018*) to identify variants derived in a genome/population of interest, H3, and ancestral in a set of other defined genomes/populations and an outgroup (i.e. multiple 'outgroups', see *Figure 3A*). The number of times the derived allele (i.e. the H3 'specific' allele) was observed in a reference individual/genome H1 was counted (nBABA$_{ex}$), as was the number of times the derived allele was observed in the target individual/genome H2 (nABBA$_{ex}$). The difference between nABBA$_{ex}$ and nBABA$_{ex}$ was then divided by the sum of nABBA$_{ex}$ and nBABA$_{ex}$, analogous to the *D* formula:

$$D_{ex} = (\text{nABBA}_{ex} - \text{nBABA}_{ex})/(\text{nABBA}_{ex} + \text{nBABA}_{ex})$$

A Z value can then be computed from bootstraps estimate of the standard error using 5 Mb genome blocks and 1000 bootstrap replicantes, to normalize the $D_{ex}$ value.

Sites must be covered at least once in each defined group ('outgroups' or otherwise). The ancestral allele was also conditioned on being at 100% frequency in each of the 'outgroup' groups, but conceptually this criteria could be slackened to investigate patterns of derived allele sharing (i.e. derived variants in H3 that are present at some frequency e.g. >0%,≤10% in one of the defined 'outgroups'). Calculations of $D_{ex}$ were based on the biallelic CpG-removed haploid-called sites (232,685,198 total), computing with and without transitions, and performed using a custom python script.

Initially, we examined the sharing of Direkli4-specific variants relative to a population of ~10,000 year old Neolithic domestic-like genomes from the Zagros Mountains (*Daly et al., 2021*), requiring the Direkli4-derived allele to be fixed for ancestral and covered at least once each in the following groups:

- Direkli bezoar
- Other bezoar (ancient and modern)
- Other *Capra*
- Sheep (using the 11 genomes defined in *Supplementary file 1C*) ultimately defining the ancestral state

This would therefore identify variants specific to the Direkli4 lineage, and exclude those shared with the other *Capra* genomes analyzed here (e.g. Caucasian Tur) or the Direkli bezoar (which have likely experienced gene flow with the Direkli4 lineage, see *Figure 2—figure supplement 4*). $D_{ex}$ values were highly similar when computed using transversions only (*Figure 3B*) or all variants (*Figure 3—figure supplement 1*, for both see *Supplementary file 1J*). To control for reference genome effects we also calculated $D_{ex}$ requiring the Direkli4-derived allele to segregate in sheep, with highly correlated results (Pearson's *r*=0.9935, *Supplementary file 1J*).

We repeated the $D_{ex}$ calculation but varying H3 to be a different genome of interest (*Figure 3—figure supplements 2 and 3*, *Supplementary file 1M*). *Capra aegagrus* from Direkli Cave (Direkli1-2, Direkli5, Direkli6) show highest levels of shared-specific ancestry, occurring in European and African goat; this ancestry is only somewhat correlated with Direkli4-specific ancestry (*r*=0.6562–0.5731) and likely reflects gene flow from Anatolian bezoar to the ancestors of west Eurasian domestic goat (e.g. *Figure 3—figure supplements 5 and 7*). A single markhor (Cfalconeri04, SAMN10736157) may have

east Asian related gene flow in its ancestry, while derived alleles signals of multiple *Capra* genomes in sub-Saharan goat imply gene flow from a *Capra* source into these domestic populations.

We observed positive $D_{ex}$ values for Direkli4 and ancient west Eurasian goats, but not modern goats (*Figure 3B*). To assess whether technical biases may artificially cause Direkli4 to share more alleles with ancient samples, we examined the correlation with $D_{ex}$ when other *Capra* genomes define the "specific" allele (*Figure 3—figure supplements 2 and 3*, *Supplementary file 1L*). Correlations of specific allele sharing values with *C. caucasica*, the Palaeolithic Armenian bezoar Hovk1, and medieval *C. cylindricornis* Caucasus1 were highest (Pearson's *r*=0.9258, 0.9231, 0.8824 respectively) (*Supplementary file 1L*). The relatively high range of correlations with historic *Capra* genomes (*r*=0.7194–0.8363) suggests some technical bias, but does not completely explain the pattern of Direkli4-specific allele sharing.

As mentioned above this 'extended *D*' approach was amenable to allowing the H3-specific variant to also segregate at a defined rate among 'outgroups', effectively 'disentangling' patterns of shared derived alleles. We investigated Direkli4-specific allele sharing, cycling through different target H2 genomes, for variants also >0%,≤10% across 'outgroups' (but fixed as ancestral in sheep). For each target H2, the total number of times a given 'Outgroup' individual shared the 'Direkli4-specific' allele was recorded, expressing as a proportion of the total >0%,≤10% shared variants in a heatmap format (*Figure 3—figure supplement 4*). While sensitive to coverage (as deeper sequenced samples will on-average make up a greater proportion of the total number of shared 'Direkli4-specific' variants), differences are apparent between Direkli4-specific allele sharing between ancient and modern European goat; the former share more Direkli4 alleles with Direkli bezoar, while the latter share more Direkli4 alleles with the Caucasian tur genomes Tur1 and Caucasus1. This pattern could be explained by a turnover in European goat populations to one with a greater Caucasian tur-related ancestry, a technical bias that increases affinity between the Direkli bezoar and ancient European goat or Tur genomes and modern European goat, or gene flow between a population related to modern European goat and Caucasian tur. These hypotheses could be tested with finer temporal sampling of European goats, or a time series of Caucasian tur to determine if gene flow from domestic populations occurred.

Python scripts to run the extended *D* calculations and to 'disentangle' the pattern of derived allele sharing are available on GitHub (*Daly, 2022*, copy archived at swh:1:rev:f803deabaa929dad5ce-beec67bb0ee3b83e3c4a9) and https://osf.io/3ecqd/.

To confirm our results were unlikely due to choice of reference genome, we recalculated a subset of $D_{ex}$ statistics with data aligned to sheep, with and without ascertaining in the sheep outgroup population and examining tests related to Direkli4-specific variants in domestic groups (*Supplementary file 1J and K*). Correlation with goat-aligned data was high (*r*=0.897 and 0.906 respectively using transversions-only), suggesting that a reference bias towards ARS1 was unlikely driving the observed patterns in Direkli4-sharing allele sharing. However, some Z scores differed by crossing the chosen significance threshold (| Z |>3) while retaining their $D_{ex}$ directionality. For example, with sheep-ascertained sites the modern European goats IOG and Italian4 obtain significant negative scores (having fewer Direkli4-specific variants than the reference Neolithic Zagros group) when using sheep-aligned data but not goat aligned (negative but not |Z|>3, *Supplementary file 1K*). This is likely due to the lower number of genomes included in the $D_{ex}$ calculation when using sheep-aligned data, a necessity due to the computational limitations of aligning available data to the sheep genome. Each additional genome include in the H4 "outgroup" (e.g. diverse bezoar, *Capra* specimen, other Direkli bezoar) should reduce the total number of sites in the $D_{ex}$ calculation, by filtering out variants otherwise assigned as 'Direkli4-specific'. Fewer sites will reduce the sensitivity of the test (by decreasing the number of sites included in each bootstrap iteration) while increasing the specificity of the variants. As such it is the consistency of directionality and correlation of the $D_{ex}$ values between different data sets which should lend confidence to the goat-aligned results presented in *Figure 3—figure supplements 1–3*.

## Graph-based modeling

To generate a graph-based admixture model of how individual genomes relate, Treemix (*Pickrell and Pritchard, 2012*) was employed on the 2X-sheep ascertained dataset. The number of migration events *m* was varied from 0 to 5, with other parameters set as -root Sheep -k 1000 --noss

--global. Node support values were estimated using 50 bootstrap replicates and the -boot option, applying node support values to the base tree using RAxML (*Stamatakis, 2014*).

To maximize the possibility of detecting gene flow between the Direkli4/Tur lineage, Orientagraph (*Molloy et al., 2021*) was employed on a reduced set of populations and at a group level. Sites were filtered to retain those with at least one call per group and a MAF of ≥0.05, leaving 104,550 sites. *m* was varied from 0 to 4 due to computational limitations, running with Treemix settings as above but with the addition of -mlno to find the Maximum Likelihood Network Orientation, and *k* of 500 due to the lower SNP number.

Finally ADMIXTOOLS2 (*Maier et al., 2022*) was employed to explore admixture graph space in a complementary manner. To investigate the admixture status of the 'taurasian tur' lineage in a limited graph space, six populations/genomes were included:

1. West Caucasus Tur *Capra caucasica* (**Tur1**).
2. Direkli4 (**DIR4**).
3. Direkli bezoar / Epipaleolithic Taurus (**ETa**).
4. PPN/PN East Iran (**NEI**).
5. Neolithic Serbia (**NSe**).
6. A group of ARS1-aligned sheep were used as the outgroup (**Sheep**).

SNPs used were the 223,772 sheep-ascertained variants described above. We followed the approach suggested by *Maier et al., 2022*, fitting 50 graphs per complexity class (m=0–5) using find_graphs(stop_gen = 100, outpop = 'Sheep') using pre-computed $f_2$ statistics calculated over all goat autosomes and allowing missingness (extract_f2(maxmiss = 1, auto_only = F)). At m=2 (two admixture edges), the preponderance of graphs fit with a worst $f_4$ outlier | Z |<3.

find_graph() was rerun at m=2 for 50 iterations, with the constraint that the Neolithic Serbian group had to receive at least one admixture edge in its history, reflecting the established gene flow event from a population related to the Direkli bezoar (*Daly et al., 2018*; *Daly et al., 2021*). Duplicate graphs were removed, as were graphs with 100%/0% admixture events. The best fitting graph was compared to the remaining graphs by calculating bootstrap values for graph scores (qpgraph_resample_multi(nboot = 100) and compare_fits()$p_emp), with significantly worse fitting graphs removed; several as-good-as fitting graphs retained significant $f_4$ outlier |Z|>3.

The remaining 11 graphs were then scored for the presence of (1) admixture edges from ETa into DIR4, an indication that the taurasian tur group has bezoar admixture, and (2) admixture edges from DIR4 into ETa, indicating that Direkli bezoar have ancestry related to the taurasian tur. In a majority of graphs (6/11), ETa is modeled as having DIR4 ancestry (median 1.5% DIR4 ancestry, mean 5.2%). Only one graph models DIR4 as a mixture of an ETa related clade and Tur1 (2% for the latter). Following the methodology suggested by *Maier et al., 2022*, we examined the best fitting graph at an additional complexity class (m=3) and found ETa again to be modeled as containing DIR4 ancestry (1%), demonstrating this feature to be a robust one. These results suggest that while the data does not exclude bezoar to taurasian tur gene flow (which is implied by other results, including mtDNA lineages *Figure 2—figure supplement 1*), 'Direkli taurasian tur' to 'Direkli bezoar' admixture was of greater consequence, with more graphs indicating Direkli bezoar received taurasian tur admixture.

All 12 graphs 'as good as' the best fitting graph for m=2 (displayed in *Figure 3—figure supplement 8*) are available at https://osf.io/3ecqd/, along with the distribution of log likelihood scores for m=0–5 and the best fitting graph for m=3.

## Data, script, and code availability

Raw sequencing reads, aligned QCed final bam files, and mitochondrial fasta files have been deposited in ENA under the project accession PRJEB51668. Admixture graphs 'as good as' the best fitting graph are available at https://osf.io/3ecqd/. *Capra taurensis* has been registered under the Zoobank LSID urn:lsid:zoobank.org:act:1261A42B-B0C0-4571-87F4-8EC3B5381A88. Scripts for extended D calculation/disentangling derived allele sharing are available on GitHub and https://osf.io/3ecqd/.

## Acknowledgements

We thank Matthew Teasdale and Amelie Scheu for their advice on interpretation of results and helpful discussions. Excavations at Direkli Cave are sponsored by T.C. Kültür ve Turizm Bakanlığı. Permission

to export samples from Direkli Cave provided by T.C. Kültür Varlıkları ve Müzeler Genel Müdürlüğü and T.C. Ankara Valılığı, Il Kültür ve Turizm Müdürlüğü, Anadolu Medeniyetleri Müzesi Müdürlüğü (#70583208–160.99(06)–899). We thank the VarGoats consortia for use of the modern tur sequencing data in IBS analyses. Preprint version 5 of this manuscript has been peer-reviewed and recommended by Peer Community In Genomics (https://doi.org/10.24072/pci.genomics.100020), and we thank all of those involved for their time and expertise. Zooarchaeological work at Direkli has been supported by grants from the Office of Vice Provost for Research, Baylor University and a URC grant from the Office of the Vice Chancellor for Research at the University of North Carolina at Chapel Hill (Benjamin S Arbuckle). This work was supported by the European Research Council under the European Union's Horizon 2020 research and innovation programme (grant numbers 885729-AncestralWeave, 295729-CodeX, 295375-Persia and its Neighbours) (Kevin G Daly, Conor Rossi, Valeria Mattiangeli, Daniel G Bradley, Eberhard Sauer); and supported in part by a Grant from Science Foundation Ireland under grant number 21/PATH-S/9515 (Kevin G Daly).

## Additional information

### Funding

| Funder | Grant reference number | Author |
| --- | --- | --- |
| University of North Carolina at Chapel Hill | URC grant from the Office of the Vice Chancellor for Research | Benjamin S Arbuckle |
| H2020 European Research Council | 885729-AncestralWeave | Kevin G Daly<br>Conor Rossi<br>Valeria Mattiangeli<br>Daniel G Bradley |
| H2020 European Research Council | 295729-CodeX | Kevin G Daly<br>Conor Rossi<br>Valeria Mattiangeli<br>Daniel G Bradley |
| H2020 European Research Council | 295375-Persia | Eberhard Sauer |
| Science Foundation Ireland | 21/PATH-S/9515 | Kevin G Daly |
| Baylor University | Office of Vice Provost for Research grants | Benjamin S Arbuckle |

The funders had no role in study design, data collection and interpretation, or the decision to submit the work for publication.

### Author contributions

Kevin G Daly, Conceptualization, Data curation, Software, Formal analysis, Investigation, Visualization, Methodology, Writing – original draft, Writing – review and editing; Benjamin S Arbuckle, Conceptualization, Resources, Formal analysis, Investigation, Visualization, Methodology, Writing – original draft, Writing – review and editing; Conor Rossi, Valeria Mattiangeli, Phoebe A Lawlor, Investigation, Writing – review and editing; Marjan Mashkour, Eberhard Sauer, Joséphine Lesur, Levent Atici, Cevdet Merih Erek, Resources, Investigation, Writing – review and editing; Daniel G Bradley, Conceptualization, Supervision, Funding acquisition, Writing – original draft, Writing – review and editing

### Author ORCIDs

Kevin G Daly (iD) http://orcid.org/0000-0002-5579-6144
Conor Rossi (iD) http://orcid.org/0000-0003-4561-8878
Valeria Mattiangeli (iD) http://orcid.org/0000-0001-9785-1714
Marjan Mashkour (iD) http://orcid.org/0000-0003-3630-9459
Levent Atici (iD) http://orcid.org/0000-0002-4929-173X
Cevdet Merih Erek (iD) http://orcid.org/0000-0002-0259-5111

Decision letter and Author response
Decision letter https://doi.org/10.7554/eLife.82984.sa2
Author response https://doi.org/10.7554/eLife.82984.sa1

## Additional files

### Supplementary files

• Supplementary file 1. Additional data tables.
 (A) Sample information. (B) Bias of D statistic tests involving Direkli4. (C) Published samples used in the present study. (D) Sample sequencing statistics. (E) Direkli1-2 (bezoar)-Direkli4 (tur-like) relative affinity measured using D(Direkli1-2, Direkli4; X, Sheep). (F) Published mtDNA sequences included in ML phylogeny. (G) Direkli4-Abdul4 relative affinity measured using D(X, Abdul4; Direkli4, Sheep). (H) Ganjdareh3-Direkli4 relative affinity measured using D(X, Ganjdareh3; Direkli4, Sheep). (I) Direkli4-Tur1 (west Caucasian tur) relative affinity measured using D(Direkli4, Tur1; X, Sheep). (J) Extended D statistic, Direkli4 set as H3 ("Direkli4-specific alleles"). (L) Extended D statistic, Direkli4 set as H3, sheep-ascertained. (M) Correlation of extended D values. (N) Extended D values, rotating H3 among different wild Capra genomes.

• Transparent reporting form

### Data availability

Raw sequencing reads, aligned QCed final bam files, and mitochondrial fasta files have been deposited in ENA under the project accession PRJEB51668. Admixture graphs "as good as" the best fitting graph are available at https://osf.io/3ecqd/. Capra taurensis has been registered under the Zoobank LSID urn:lsid:zoobank.org:act:1261A42B-B0C0-4571-87F4-8EC3B5381A88. Scripts for extended D calculation/disentangling derived allele sharing are available on GitHub, (copy archived at swh:1:rev:f803deabaa929dad5cebeec67bb0ee3b83e3c4a9).

The following datasets were generated:

| Author(s) | Year | Dataset title | Dataset URL | Database and Identifier |
|---|---|---|---|---|
| Daly KG | 2022 | A novel lineage of the Capra genus discovered in the Taurus Mountains using ancient genomics | https://www.ebi.ac.uk/ena/browser/view/PRJEB51668 | European Nucleotide Archive, PRJEB51668 |
| Daly KG | 2022 | Direkli Cave Capra lineage | https://osf.io/3ecqd/ | Open Science Framework, 3ecqd |
| Daly KG | 2022 | Zoobank LSID of Taurensis Tur | https://zoobank.org/NomenclaturalActs/5c272cd4-f4a8-4694-a0de-4ae9a56b8349 | zoobank, urn:lsid:zoobank.org:act:5C272CD4-F4A8-4694-A0DE-4AE9A56B8349 |

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
