## [Decision Letter]

[Editors' note: This article has been reviewed and recommended by Peer Community in Genomics (https://doi.org/10.24072/pci.genomics.100020).]

---

## [Author Response]

**Evaluation summary:**

Complex histories of human-animal interactions – involving long-standing genetic variation, population structure, local adaptation, climate change, hunting pressures, domestication, and admixture/introgression – have repeatedly played out worldwide. By studying these processes, we can advance our understanding of past human behavior, the ecological and evolutionary fates of animals affected by human- and climate-based changes to their environment, and the biological origins of our present-day domesticates and any of their still-extant wild relatives. Here, Daly et al. generate both ancient (samples as old as 14,000 years before present) and historical DNA genomic sequence data to provide a valuable case study of such intersecting phenomena in the history of the genus Capra (goat). The presented results are convincing, and interesting alongside those available for horses (with distinct economic roles and generational travel distances for goats and horses). Comparisons with future datasets from other taxa are also eagerly anticipated. As part of this work, the authors also develop a modified D statistic, Dex, to help study gene flow in complex ancestry scenarios – this statistic will likely be useful in other contexts.